# Cardiac Arrhythmias Requiring Electric Countershock during the Neonatal Period—A Systematic Review

**DOI:** 10.3390/children10050838

**Published:** 2023-05-05

**Authors:** Nathalie Oeffl, Marlies Krainer, Stefan Kurath-Koller, Martin Koestenberger, Bernhard Schwaberger, Berndt Urlesberger, Lukas P. Mileder

**Affiliations:** 1Division of Pediatric Cardiology, Department of Pediatrics and Adolescent Medicine, Medical University of Graz, 8036 Graz, Austria; stefan.kurath@medunigraz.at (S.K.-K.);; 2Division of Neonatology, Department of Pediatrics and Adolescent Medicine, Medical University of Graz, 8036 Graz, Austria; marlies.krainer@medunigraz.at (M.K.); lukas.mileder@medunigraz.at (L.P.M.)

**Keywords:** neonate, preterm, atrial flutter, ventricular tachycardia, supraventricular tachycardia, ventricular fibrillation, cardiac arrest, electric countershock, defibrillation

## Abstract

Background: In neonates, cardiac arrhythmias are rare. Electric countershock therapy is an effective alternative to drug therapy for neonatal arrhythmias. There are no randomized controlled studies investigating electric countershock therapy in neonates. Objective: To identify all studies and publications describing electric countershock therapy (including defibrillation, cardioversion, and pacing) in newborn infants within 28 days after birth, and to provide a comprehensive review of this treatment modality and associated outcomes. Methods: For this systematic review we searched MEDLINE, EMBASE, Cochrane Central Register of Controlled Trials (CENTRAL), and Cumulative Index to Nursing and Allied Health Literature (CINAHL). All articles reporting electric countershock therapy in newborn infants within 28 days after birth were included. Results: In terms of figures, 113 neonates who received electric countershock due to arrhythmias were reported. Atrial flutter (76.1%) was the most common arrhythmia, followed by supraventricular tachycardia (13.3%). Others were ventricular tachycardia (9.7%) and torsade de pointes (0.9%). The main type of electric countershock therapy was synchronized cardioversion (79.6%). Transesophageal pacing was used in twenty neonates (17.7%), and defibrillation was used in five neonates (4.4%). Conclusion: Electric countershock therapy is an effective treatment option in the neonatal period. In atrial flutter especially, excellent outcomes are reported with direct synchronized electric cardioversion.

## 1. Introduction

Cardiac arrhythmia in newborn infants is rare. The incidence varies between 1 and 5% of neonates who are admitted to neonatal intensive care units. These cases also include benign arrhythmias, such as premature atrial contractions and premature ventricular contractions [1], which are more common in term compared to preterm infants [1].

The neonatal heart rate normally ranges between 100 and 160 beats per minute (bpm) after completed neonatal transition [2]. A long-lasting heart rate above 180 bpm can result in a medical emergency due to hemodynamic compromise and heart failure [3]. Cardiac arrhythmias are problematic in newborn infants because significant changes in heart rate may result in a decline in cardiac output followed by impaired filling and venous congestion [3]. In preterm infants especially, due to immaturity, cardiac arrhythmias can lead to hemodynamic decompensation within a short time [3].

The most common arrhythmias during the newborn period are as follows: (i) atrial flutter (AFL) (Figure 1), (ii) supraventricular tachycardia (SVT), mostly due to atrioventricular re-entrant tachycardia (Figure 2), and (iii) ventricular tachycardia (VT) [4]. Atrioventricular arrhythmias account for 90% of neonatal tachyarrhythmias, whereas VT is uncommon and mostly caused by other factors, including viral infection, Ro-antibody-mediated myocarditis, structural heart diseases, or long QT syndrome [3,4].

Electric countershock therapy includes defibrillation (DF), transesophageal pacing (TEP), and synchronized electric cardioversion (CV). DF is an unsynchronized electric countershock primarily used to restore a cardiac rhythm during cardiac arrest and cardio-pulmonary resuscitation [5]. Transesophageal overdrive pacing and synchronized electrical CV are used in the treatment of atrioventricular tachyarrhythmias to restore the sinus rhythm [6]. In the former, catheter placement is needed and success rates range from only 30% to about 70% for converting AFL into the sinus rhythm [6,7,8,9]. The latter is an electrocardiogram-synchronized electric countershock therapy, which has been shown to be the most successful method with which to treat AFL, with success rates above 85% [6,7,8,9,10].

However, there are no randomized controlled studies, very limited data, and no international guidelines reporting treatment options and outcomes of newborn infants with cardiac tachyarrhythmia. The aim of this review was to perform a systematic search of available literature and to summarize the interventions and outcomes of cardiac arrhythmia treated with electric countershock in newborn infants aged ≤ 28 days.

## 2. Materials and Methods

For this review, the stepwise approach specified in the Preferred Reporting Items for Systematic Reviews and Meta-Analysis (PRISMA) statement was used. The systematic review was registered in PROSPERO (CRD42022370306) Appendix A.

### 2.1. Search Strategy and Inclusion Criteria

We searched the following databases: MEDLINE (through PubMed), EMBASE, Cochrane Central Register of Controlled Trials (CENTRAL), and Cumulative Index to Nursing and Allied Health Literature (CINAHL) from their inception to the 29 January, 2023. For the systematic electronic search, we used the MeSH terms “infant”, “newborn”, and “electric countershock”. Additional publications were identified through a manual search of references in the retrieved original manuscripts and review articles.

Peer-reviewed articles, including case reports, case series, and studies, were eligible for inclusion if they described electric countershock therapy (including DF, CV, and TEP) in newborn infants with a postnatal age of ≤28 days. We excluded animal studies, unpublished studies, reports, and studies published as abstracts only, in addition to studies that did not describe electric countershock therapy (e.g., medical CV only). Furthermore, we excluded articles written in languages other than English or German and those articles where the full texts were not available.

### 2.2. Study Selection

The results of the literature search were uploaded onto the COVIDence collaboration tool (https://www.COVIDence.org/ accessed on 30 January 2023; COVIDence, Melbourne, Australia) and duplicates were removed automatically. Two reviewers (N.O. and M.K.) independently screened all of the study titles and abstracts for eligibility. This was followed by an independent full-text assessment performed by the same reviewers (N.O. and M.K.). Conflicts were discussed and resolved with a third member of the review team (L.P.M.). Authors were excluded from voting if they were an author of the study or case report.

### 2.3. Data Extraction and Charting

Data were extracted by one reviewer (N.O.) via the use of the COVIDence collaboration tool and Microsoft Excel (Microsoft, Redmond, WA, USA), and discrepancies in the data extraction were resolved with another reviewer (M.K.). The following data were extracted: authors, year of publication, number of included neonates, gestational age, sex, congenital heart defects, age at electric countershock therapy, electrocardiographic diagnosis, prenatal diseases, medication before electric countershock, effective treatment (defined as the successful cessation of the cardiac arrhythmia), and recurrence during follow-up.

## 3. Results

The primary literature search yielded 181 articles. After removing duplicates, title and abstract screening was performed; 77 publications were assessed for full-text reviews. Finally, 39 articles met the inclusion criteria (Figure 3). The study details are summarized in Table 1.

The reviewed articles included 27 case reports, 2 prospective studies, and 10 retrospective studies. In summary, 113 neonates, who received an electric countershock due to arrhythmias, were reported. Of these, only two neonates (1.8%) were shocked in an outpatient setting [11,12].

### 3.1. Gestational Age and Gender Distribution

Thirty-four articles (87.2%) [11,12,13,14,15,16,17,18,19,20,21,22,23,24,25,26,27,28,29,30,31,32,33,34,35,36,37,38,39,40,41,42,43,44] stated the patients’ gestational ages. The described neonates ranged from 25^+2^ to 40^+4^ weeks of gestation. One case [13] reported electric CV in an extremely preterm infant (25^+2^ weeks of gestation), and another study [14] in a very preterm infant (28^+0^ weeks of gestation). In twenty-five articles (64.1%) [7,8,12,13,14,15,16,17,18,19,20,21,22,23,24,31,35,36,37,38,41,42,43,44,45] including sixty-one patients (54%), the patients’ sex was described. In these articles, females (*n* = 29, 47.5%) and males (*n* = 32, 52.5%) were equally affected by arrhythmias.

### 3.2. Age at Electric Countershock

In 22 articles (56.4%) [7,13,14,15,16,17,18,19,20,21,22,23,24,25,26,27,28,29,30,31,32,33] including 39 neonates (34.5%), rhythm disturbances were diagnosed immediately or within the first hours after birth. In 6 articles (15.4%) [8,34,35,36,37,38] the 62 described neonates (54.9%) were classified into an age group between birth and four weeks post-partum. Eleven articles (28.2%) [11,12,39,40,41,42,43,44,45,46,47] including twelve neonates (10.6%) reported on the manifestation of arrhythmias between the second and the twenty-fourth day after birth. In two cases arrhythmias were due to surgery [41,43], and in two cases they occurred during the placement of a central venous catheter [25,40].

In 15 studies (38.5%) [7,8,14,15,16,17,20,22,23,26,28,29,32,34,35] including 33 neonates (29.2%), arrhythmias had already been detected prenatally or were the reason for inducing labor [22,23].

### 3.3. Congenital Heart Disease

Six articles (15.4%) [14,22,35,36,38,39] including seven neonates (6.2%) reported on arrhythmias accompanying congenital heart disease, including the transposition of the great arteries (*n* = 2) [35,38], Ebstein’s disease (*n* = 1) [14], and a univentricular heart (*n* = 1) [38]. One article [38] reported on arrhythmias in neonates related to the transposition of the great arteries and a univentricular heart, but did not offer exact numbers of affected patients in the included age group.

In two case reports [19,20], arrhythmia was related to pulmonary hypertension. Arrhythmia in newborn infants with congenital heart disease appeared preoperatively [14] as well as after correction or surgical palliation [41]. In one case report, the first manifestation of fetal AFL at 31 weeks of gestational age was in a neonate with Ebstein’s disease [14].

### 3.4. Arrhythmias

AFL was described in 86 neonates (76.1%) with arrhythmias, who received electric countershock. In 12/86 neonates (14%), the typical ECG pattern of AFL had been revealed after using adenosine [13,16,19,22,23,31,47]. 

SVT was the second most common arrhythmia during the neonatal period. In total, 15 of the included neonates (13.3%) suffered from SVT. Causes for SVT were idiopathic [24,30], complications during central venous catheter placement [40], and pulmonary hypertension [20].

In addition to these two most common arrythmias, only a few other forms were described. In 11 neonates (9.7%) VT appeared to be related to asphyxia [18] or surgery [41,43]. One (0.9%) of the included infants suffered from torsade de pointes tachycardia due to long QT Type 3 syndrome [21]. One asphyctic neonate [33] with postnatal ventricular fibrillation (VF) had a mutation in the SCN5A gene, which is present in 5–10% of patients with long QT syndrome.

### 3.5. Type of Electric Countershock

In 16 studies (41%) [8,12,14,15,16,18,19,20,25,32,33,34,35,39,41,43] including 49 patients (43.4%), direct (primary) synchronized electric CV was used. Nineteen studies (48.7%) [7,13,17,18,21,22,23,24,26,27,28,30,31,36,37,38,40,42,45] including forty-seven patients (41.6%) described antiarrhythmic premedication, including betablockers and digoxin, for heart rate control or adenosine for revealing the underlying arrhythmia. In 4 articles (10.3%) [11,29,46,47] including 17 patients (15%) no information about antiarrhythmic premedication was available.

The main type of electric countershock therapy was synchronized electric CV in 90 neonates (79.6%). In 43 neonates (47.8%) receiving synchronized electric cardioversions, direct synchronized electric CV was successful, regardless of the type of arrhythmia. Only one case [28] reported AFL that was refractory to both antiarrhythmic medication and synchronized electric CV. In this case, the sinus rhythm could be restored by the continuous administration of intravenous flecainide.

**Table 1 children-10-00838-t001:** Demographic data of the included studies.

Publication	Year	Study Design	Patient(-s) (*n* = )	Age at ES (Days)	Congenital Heart Defect	Gestational Age (Weeks)	ECG Diagnosis (*n* = )	Arrhythmia Prenatally Detected	Medication before ES (*n* = )	Success	Energy Level	Recurrence during Follow-Up (*n* = )
Barclay et al. [39]	1972	Case report	1	10	ASD	≥37	AFL	No	Digoxin	CV: yes	n/a	No
Heinonen et al. [18]	1992	Case report	1	0	No	≥37	VT	No	No	CV: yes	1 J/kg	No
Casey et al. [34]	1997	Retrospective study	18	0–30	n/a	30–38	AFL (*n* = 16) SVT (*n* = 2)	Yes (*n* = 3)	Digoxin (*n* = 16)	CV: *n* = 16 TEP: *n* = 2	n/a	No (*n* = 18)
Allegaert et al. [17]	2002	Case report	1	0	No	38 + 0	AFL	Yes	No	CV: yes	1 × 1 J/kg 1 × 1.5 J/kg	No
Cornwell et al. [16]	2005	Case report	2	0	No	34–40	AFL (*n* = 2)	Yes (*n* = 2)	Adenosine (*n* = 2)	CV: yes (*n* = 2)	1 J/kg 2 J/kg	No (*n* = 2)
Ceresnak et al. [35]	2009	Retrospective study	6	0–14	TGA (*n* = 1) pulmonary atresia + intact IVS (*n* = 1)	34–40	AFL (*n* = 6)	Yes (*n* = 4)	No	CV: yes	1 J/kg	Yes *n* = 2
Silva et al. [40]	2010	Case report	1	14	No	≥37	SVT	No	No	CV: yes	1 J/kg	No
Gulletta et al. [15]	2011	Case report	1	0	n/a	≥37	AFL	Yes	Adenosine Propafenone	CV: yes	1 J/kg	No
Rein et al. [24]	1986	Case report	1	0	No	33	SVT	No	Digoxin	CV: yes	3 J/kg	Yes
Lisowski et al. [7]	1999	Retrospective study	9	0	n/a	34–40	AFL (*n* = 9)	Yes (*n* = 9)	Yes	CV: yes (*n* = 9)	n/a	n/a (*n* = 9)
Paech et al. [21]	2001	Case report	1	0	No	≥37	TdP	No	no	CV: yes	2 J/kg	No
Sinha et al. [25]	2005	Case report	1	0	No	≥37	AFL	No	no	CV: yes	2 × 0.5 J/kg 1 × 1 J/kg	No
Konak et al. [19]	2013	Case report	1	0	Pulmonary hypertension	34 + 0	AFL	No	Adenosine Amiodarone	CV: yes	1.4 J/kg	Yes
Nijres et al. [20]	2015	Case report	1	0	Pulmonary hypertension	≥37	SVT	Yes	Adenosine	CV: yes	2 J/kg	Yes
Poryo et al. [22]	2017	Case report	1	0	Hypertrophic cardiomyopathy	34 + 3	AFL	Yes	Adenosine	CV yes	1.4 J/kg	No
Apostolidou et al. [23]	2018	Case report	1	0	No	≥37	AFL	Yes	Adenosine	CV: yes	1 × 1 J/kg 2 × 1.5 J/kg	No
Theodorou et al. [41]	2003	Case report	1	10	Truncus arteriosus	≥37	VF	No	No	DF: no	1 × 3 J/kg 5 × 6 J/kg	Died
Umeh et al. [13]	2017	Case report	1	0	no	25 + 2	AF	No	Adenosine	CV: yes	1 J/kg	No
Vintzileos et al. [26]	1986	Case report	1	0	no	33	AFL	Yes	Digoxin	CV: yes	1.7 J/kg	Died
Hassenrück et al. [27]	1965	Case report	1	0	no	36	AFL	No	Digoxin	CV: yes	n/a	No
Liberman et al. [42]	2006	Prospective study	1	14	no	n/a	AFL	No	n/a	CV: yes	0.5 J/kg	No
Mehta et al. [36]	1993	Retrospective study	3	0–4	Interrupted arch + VSD (*n* = 1)	n/a	AFL (*n* = 3)	No	Digoxin (*n* = 3)	CV: yes (*n* = 3)	0.25 J/kg	n/a (*n* = 3)
Miyake et al. [11]	2013	Retrospective study	1	14	no	n/a	VF	No	n/a	DF: yes	n/a	Yes
Peng et al. [46]	1998	Retrospective study	2	1–3	No	≥37	AFL (*n* = 2)	No	No	CV: yes (*n* = 2)	n/a	No (*n* = 2)
Roumiantsev et al. [47]	2017	Retrospective study	1	2	No	36 + 3	AFL	No	Adenosine Esmolol	CV: yes	0.5 J/kg	No
Saidi et al. [43]	1998	Case report	1	9	No	≥37	VT	No	Adenosine Amiodarone Lidocaine	CV: yes	2 × 1.5 J/kg 1 × 1.5 J/kg	Yes
Singh et al. [44]	1989	Case report	1	24	No	≥37	SVT	No	No	CV: yes	20 J *	Yes
Sugrue et al. [45]	1985	Case report	2	9–21	No	≥37	SVT (*n* = 2)	No	No	CV: yes (*n* = 2)	2 J/kg	Yes (*n* = 2)
Suzumura et al. [28]	2004	Case report	1	0	No	≥37	AFL	Yes	No	TEP: no CV: no	2 J/kg	No
Takei et al. [29]	2015	Case report	1	0	No	37 + 0	AFL	Yes	No	CV: yes	n/a	No
Texter et al. [8]	2006	Retrospective study	20	0–11	No	≥37	AFL (*n* = 20)	Yes (*n* = 3)	No	CV (*n* = 20) TEP (*n* = 7)	n/a	Yes (*n* = 5)
Tunca Sahin et al. [14]	2021	Retrospective study	6	0	Ebstein (*n* = 1)	28–41	AFL (*n* = 6)	Yes (*n* = 6)	No	CV: yes (*n* = 6)	n/a	Yes (*n* = 4)
Wells et al. [30]	1987	Case report	1	0	No	≥37	SVT	No	No	CV: yes	5 J/kg	No
Wójtowicz-Marzec et al. [31]	2020	Case report	1	0	No	34 + 0	AFL	No	Adenosine	CV: yes	1 J/kg	No
Yilmaz-Semerci et al. [32]	2018	Case report	3	0	no	34–38	AFL (*n* = 2) SVT (*n* = 1)	Yes (*n* = 1)	Propanolol (*n* = 2) Adenosine (*n* = 1)	CV: yes (*n* = 3)	1 J/kg	Yes (*n* = 3)
Tejman-Yarden et al. [12]	2017	Case report	1	14	no	≥37	VT	No	No	DF: yes	60 J/kg	No
Tibballs et al. [37]	2011	Prospective observation Study	5	<30	n/a	n/a	VT VF	n/a	n/a	DF: yes (*n* = 5)	0.5–3 J/kg	n/a (*n* = 5)
Dick et al. [38]	1988	Retrospective study	10	<30	TGA UVH	n/a	AFL SVT	n/a	n/a	TEP: yes (*n* = 10)	n/a	n/a (*n* = 10)
Mileder et al. [33]	2021	Case report	1	0	no	≥37	VT	No	Epinephrine Calcium Magnesium	DF: yes	1.4 J/kg	No

AFL: atrial flutter; AFIB: atrial fibrillation; ASD: atrial septum defect; AT: atrial tachycardia; CV: electric cardioversion; DF: defibrillation; ES: electric countershock; IVS: interventricular septum; n/a: not available; SVT: supraventricular tachycardia; Tdp: torsade de pointes, TEP: transesophageal pacing; TGA: transposition of the great arteries; UVH: univentricular heart; VF: ventricular fibrillation; VT: ventricular tachycardia; and * cumulative shock energy (no weight available).

TEP was performed in twenty neonates (17.7%) [8,28,34,38]. Dick et al. [38] reported good outcomes in atrial overdrive pacing, while it was reported as being unsuccessful in other publications [8,28,34].

DF was used in five newborn infants (4.4%), all during cardiopulmonary resuscitation [11,12,33,37,41].

### 3.6. Energy Levels for Electric Countershock

The energy used for synchronized electric CV varied between 0.25 J/kg and 3 J/kg. The highest energy level for the first attempt was 1 J/kg. If this was not successful, the energy level of the subsequent attempt was usually doubled.

### 3.7. Outcome and Recurrence

The range of follow-ups varied from two months to sixteen years (median of twelve months). In all of the included articles, two deaths (1.8%) were reported. One patient [41] died due to VF following the surgical correction of truncus arteriosus type II, the other because of congestive heart failure [26].

In 23 (20.3%) out of the 113 included patients a relapse of arrhythmia was reported. The characteristics of patients with arrhythmia relapse after electric countershock therapy are summarized in Table 2.

The time from countershock until relapse ranged from 1 h up to a maximum of 72 h. There was no difference in the recurrence rate, whether the initial therapy was direct electric CV or electric CV after antiarrhythmic medication. Twenty (86.9%) out of twenty-three neonates with relapse received at least one prophylactic antiarrhythmic medication after the relapse episode, including beta-blockers and Vaughan-Williams class I and II agents, or digoxin. None of these patients relapsed within the mentioned follow-up periods. One neonate underwent left cardiac sympathetic denervation and the placement of an epicardial defibrillator due to recurrent VT [11].

## 4. Discussion

In this systematic review we focused on arrhythmias occurring in the postnatal period and being treated by electric countershock. Arrythmias at this age are rare, and treatment is often challenging. Arrhythmias may have severe consequences, such as congestive heart failure or even death when untreated [7,34]. On the other hand, if treatment is successful, excellent outcomes are reported [34]. The reason why sustained tachycardia results in heart failure has not been fully clarified yet, but reduced coronary blood flow with subsequent subendocardial ischemia has been discussed [48]. Electric countershock therapy is an effective alternative to drug therapy and well proven in children. The first successful direct electric CV of a neonate with atrial flutter was described almost 60 years ago [27].

Due to fetal echocardiography and the improvement of M-mode as well as Doppler techniques, sustained fetal arrythmias are now diagnosed and treated at early stages [49]. In small series [8] especially, AFL occurred shortly after birth or within the first 48 h of postnatal life. We found that almost half of them (40%) were detected prenatally. In such a condition, developing a strategy for a newborn to restore the sinus rhythm quickly after birth is essential.

In fetal arrythmias especially, a multidisciplinary management plan developed by gynecologists, pediatric cardiologists, and neonatologists is required. The most serious complication of sustained fetal arrythmias is the development of congestive heart failure with hydrops fetalis [50]. This condition has a great impact on pre- and postnatal care, especially in regard to the choice of antiarrhythmic treatment and the response of therapy. It also results in a higher postnatal morbidity and mortality compared to neonates without a history of hydrops fetalis [50].

The most commonly reported arrythmia treated by CV was AFL. It is defined as a re-entry tachycardia originating in the atrium with heart rates between 250 and 350 bpm. In its common form, AFL circles clockwise or counterclockwise within the right atrium using the cavo-tricuspid isthmus as a critical structure [51]. The typical ECG appearance of sawtooth flutter waves (F-waves) depends on the atrioventricular conduction rate, which frequently varies between 1:1 and 4:1 [52]. Because there is no involvement of the AV node, adenosine may support diagnoses by inducing AV block and revealing F-waves.

Small series [7,8] postulated direct CV as the most effective method in the treatment of AFL. Texter et al. [8] proposed a waiting period of 6 to 12 h in asymptomatic patients to increase the chance of spontaneous conversions. Nonetheless, there are currently no data with regard to the spontaneous conversion rates of AFL in this age group.

Several studies [8,34,46] found no association between AFL and congenital heart disease. For SVT, on the other hand, associations with all types of congenital heart disease have been described, especially in Ebstein’s disease or the L-transposition of the great arteries [53]. Therefore, echocardiography should be performed in every neonate with arrhythmias to exclude congenital heart disease and assess ventricular function. The impairment of left ventricular function is mainly related to the duration of tachycardia [8].

Aside from congenital heart disease, other risk factors for SVT, including electrolyte disturbances [53], sepsis [53], or the placement of central venous catheters [40], have been described. SVT without cardiac anomalies in neonates is often due to atrioventricular re-entry tachycardia accelerated by an accessory pathway. Due to the pathophysiological involvement of the AV node, vagal maneuvers and intravenous adenosine are the first treatment modes of choice. Synchronized direct CV should be limited to hemodynamically compromised infants or refractory SVT [51].

Regarding the performance of CV, the European Resuscitation Council recommends a starting energy level of 1 J/kg for pediatric patients. In the case of failure, the energy of the subsequent attempts should be doubled, escalating up to 4 J/kg [54]. In the literature, success with lower energy (0.2–0.5 J/kg) has been reported [42]; however, one study [35] reported on elevated impedance during CV for AFL using neonatal shock pads, underlining the importance of adequate energy levels.

Reported complications following CV treatment were skin burns, hypotension, and pulmonary edema. In our review, no major consequences were reported, not even in extremely premature neonates [13].

In neonates, the experience with TEP is limited to a few case reports [28,45] and small case series [8,38]. Texter et al. [8] and Dick et al. [38] described TEP as a safe and effective method in restoring the sinus rhythm in this age group. It has also been mentioned to be a temporary treatment option in refractory arrhythmias [45]. Future studies will be necessary to evaluate the effectiveness of TEP in contrast to other types of electric countershock in infancy.

DF is rarely required for cardiopulmonary resuscitation at birth or during the neonatal period. While DF is discussed in detail in advanced pediatric life support guidelines [54], it is not even mentioned in recent neonatal resuscitation guidelines [55]. Shockable rhythms, such as VT and VF, in this age group are mainly associated with open-heart surgery [41], inherited channelopathies [21,33], or coronary anomalies [56]. Manual DF is the method of choice [54], but automatic external defibrillators have also been successfully used [12].

In AFL without associated congenital heart disease especially, relapse rates of under 10% after initial treatment are described [8]. In the case of relapse, episodes predominantly occurred within the first 72 h after the initial electric countershock. No study described a late relapse. In AFL without relapse episodes, no prophylactic antiarrhythmic treatment is currently being recommended [52]. In contrast, patients with SVT need a closer follow-up strategy. A recent study describes 460 infants with SVT and proposes Wolff–Parkinson–White syndrome, fetal or late diagnosis (>60 days), and multi-antiarrhythmic therapy as the main clinical predictors for relapse [57]. If prophylactic antiarrhythmic medication is necessary, for example, as in relapsing arrhythmias, the exact choice of medication as well as the duration of prophylactic therapy for AFL or SVT differ among studies (Table 2) and remain heterogenous [58].

Arrhythmias in congenital heart disease or following heart surgery should be considered contrary to the above-mentioned factors. These are associated with high morbidity and mortality [59].

Our findings suggest that, even though electric countershock is a rare intervention in neonates, every neonatal intensive care unit should be equipped with a defibrillator which staff must be trained regularly on how to use in case of cardiac arrhythmia, especially when dealing with patients with a history of fetal arrhythmia.

## 5. Conclusions

In our systematic literature review we found that electric countershock therapy is an effective treatment option in the neonatal period, with low numbers of reported complications and low relapse rates. Especially in neonates with AFL, excellent outcomes with direct synchronized electric CV were reported.

## 6. Limitation

A limitation of our reviewed data is a possible reporting bias, since most included studies were case reports and case series with favorable success rates.

## Figures and Tables

**Figure 1 children-10-00838-f001:**
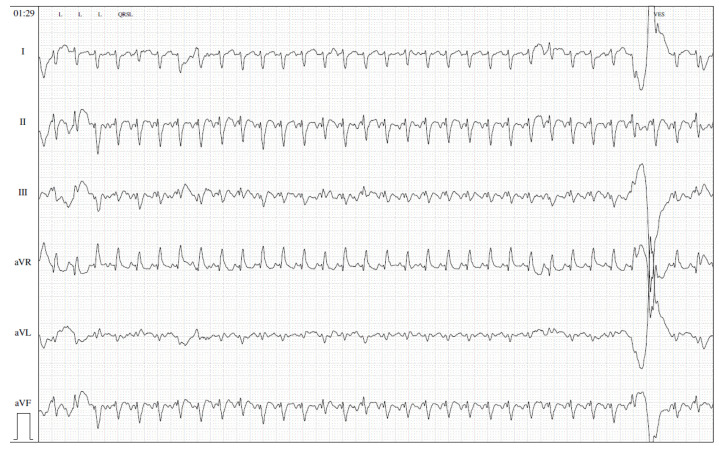
Atrial flutter in a 3-week-old boy with a surgically corrected coarctation (50 mm/s).

**Figure 2 children-10-00838-f002:**
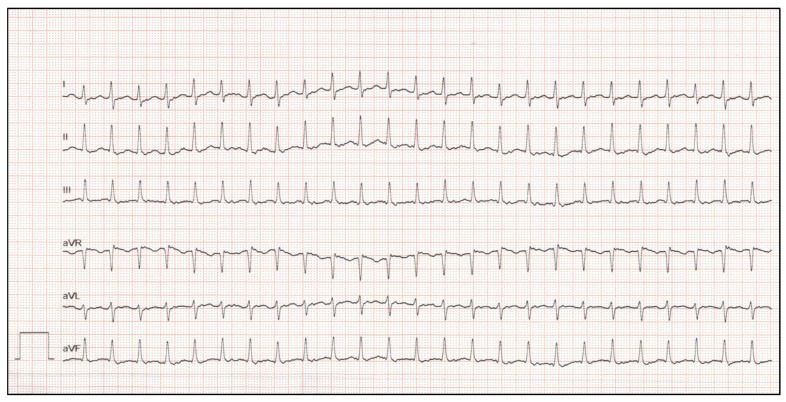
Supraventricular tachycardia in a 2-week-old boy with a heart rate of 304 bpm without congenital heart disease (50 mm/s).

**Figure 3 children-10-00838-f003:**
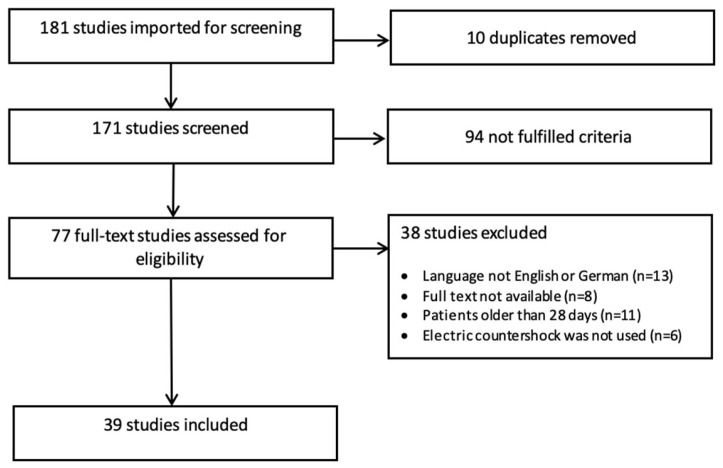
PRISMA (Preferred Reporting Items for Systematic Reviews and Meta-Analysis) flow diagram of the study selection.

**Table 2 children-10-00838-t002:** Characteristics of patients with relapse of arrhythmia after electric countershock.

Patient	Publication	ECG Diagnosis	Medication before ES	ES Successful	Energy Level	Total Number of Shocks (*n*)	Recurrence after First Shock (Hours)	ES Repeats (*n*)	Energy Level	Medication after ES	Follow-Up Period
1	Ceresnak et al. [35]	AFL	No	CV: yes	1.5 J/kg	n/a	n/a	n/a	n/a	Digoxin	n/a
2	Ceresnak et al. [35]	AFL	No	CV: yes	1.5 J/kg	n/a	n/a	n/a	n/a	Digoxin	n/a
3	Konak et al. [19]	AFL	Amiodarone	CV: yes	1.4 J/kg	*n* = 1	6–48	*n* = 4	1.4 J/kg	Propafenone	3 months
4	Miyake et al. [11]	VF	Amiodarone	DF: yes	n/a	n/a	2	*n* = 5	n/a	Amiodarone Propanolol	17 months
5	Saidi et al. [43]	VT	Adenosine Amiodarone Lidocaine	CV: yes	1.5 J/kg	*n* = 2	24	*n* = 1	2 J/kg	Amiodarone	n/a
6	Singh et al. [44]	SVT	No	CV: yes	n/a	*n* = 1	12	*n* = 1	n/a	Digoxin	9 months
7	Singh et al. [44]	SVT	No	CV: yes	20 J *	*n* = 1	12	*n* = 1	10 J *	Digoxin	n/a
8	Sugrue et al. [45]	SVT	No	CV: yes	2 J/kg	1	<12	*n* = 6	2 J/kg	Digoxin Quinidine	2 months
9	Sugrue et al. [45]	SVT	No	CV: yes	2 J/kg	1	<72	*n* = 10	2 J/kg	Digoxin Verapamil	3 months
10	Texter et al. [8]	AFL	No	TEP: yes	n/a	n/a	n/a	n/a	n/a	Digoxin Propanolol	n/a
11	Texter et al. [8]	SVT	Flecainide Amiodarone	TEP: no	n/a	n/a	n/a	n/a	n/a	Digoxin Amiodarone	n/a
12	Texter et al. [8]	AFL	Amiodarone	TEP: no CV: no	n/a	n/a	n/a	n/a	n/a	Amiodarone Propanolol Digoxin	n/a
13	Texter et al. [8]	SVT	Procain-amide	TEP: no CV: yes	n/a	n/a	n/a	n/a	n/a	Flecainide	n/a
14	Texter et al. [8]	AFL	No	TEP: no	n/a	n/a	n/a	n/a	n/a	Flecainide Digoxin	n/a
15	Tunca Sahin et al. [14]	AFL	No	CV: yes	n/a	n/a	<6	n/a	n/a	No	13 years
16	Tunca Sahin et al. [14]	AFL	No	CV: yes	n/a	n/a	<6	n/a	n/a	Flecainide Atenolol Digoxin	16 years
17	Yilmaz-Semerci et al. [32]	AFL	Propanolol	CV: yes	1 J/kg	*n* = 1	36–48	*n* = 1	2 J/kg	Sotalol	12 months
18	Yilmaz-Semerci et al. [32]	SVT	Adenosine	CV: yes	n/a	*n* = 1	3	*n* = 2	n/a	Propanolol	12 months
19	Yilmaz-Semerci et al. [32]	AFL	Adenosine, propanolol	CV: yes	n/a	*n* = 1	<1	n/a	n/a	Amiodarone Propanolol	12 months
20	Tunca Sahin et al. [14]	AFL	No	CV: yes	n/a	n/a	<6	n/a	n/a	Flecainide Digoxin	10 years
21	Tunca Sahin et al. [14]	AFL	No	CV: yes	n/a	n/a	<6	n/a	n/a	Flecainide Amiodarone Digoxin	11 years

Remark: Two case reports [20,24] with a recurrence of arrhythmia from Table 1 are not included in Table 2, as the relapse episodes resolved spontaneously. AFL: atrial flutter; CV: electric cardioversion; ES: electric countershock; n/a: not available; SVT: supraventricular tachycardia; TEP: transesophageal pacing; VF: ventricular fibrillation; VT: ventricular tachycardia; and *cumulative shock energy (no weight available).

## Data Availability

No new data were generated.

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
