# Peer review of "Cardiac Arrhythmias Requiring Electric Countershock during the Neonatal Period—A Systematic Review"

_children, 2023, doi:10.3390/children10050838_

Round 1
Reviewer 1 Report
I would sincerely like to thank the editor for the opportunity of reviewing this research that reaches such an interesting topic, and the authors for their hard work of summarizing all the available data from literature regarding this subject. They have done a systematic research in literature, offering us an accumulation of very important information.
I would also like to congratulate them for registering their important work in PROSPERO. It is also important to acknowledge their work, the fact that literature contains no review regarding this specific topic. Giving the excellent outcomes described by studies regarding the use of electric counter shock therapy in neonates, I think it is a subject that is worth being presented. The article is thoroughly documented and presented in a easily to follow and understand manner.
Abstract section:
The abstract exceeds 200 words. Correct this aspect.
Row 22-23 from the abstract section – rephrase.
Introduction section:
Row 117 – “therapy success” – rephrase.
Results section:
Row 122 – “and 77 publications” without the word “and”.
Figure 3 – rephrase – “wrong” doesn’t sound academic; and what does wrong outcome and setting refer to?
Row 131 – a comma is missing after “of these”.
Row 138 – “There was no difference in sex in the reviewed studies” – rephrase.
Rows 143-144 - “In six articles (15.4%) 143 [8,34–38], including 62 neonates (54.9%), the age of these patients was defined by a range 144 from birth to four weeks post partum.” – rephrase to explain better.
Row 155 – “arrhythmia and accompanying: without the “and”.
Row 163 – last sentence from the paragraph needs rephrasing.
Row 166 – a comma is missing before the word “who”.
Row 168 – “by the application of adenosine” – rephrase, perhaps “by the use” or “after using” would be better.
Row 176 – “QT 3 syndrome” what does “3” refers to?
Row 182 – a comma is missing after the word “cardioversion”.
Rows 183-185 – some commas missing.
Row 186 – change “on” in to “an”.
“Dick et al [38] reported 1 good outcomes in atrial overdrive pacing, while it was reported as unsuccessful in other 2 publications [8,28,34].” – this needs to be moved into the discussion section.
Row 21 – “the rate” of recurrence.
Discussion section:
Row 40 – the focus was on neonatal period, post-natal, after the birth not perinatal.
Row 47 – without the word “and”.
Row 48 – without “already”
Row 76 – a comma after “tachycardia”
Row 89 – a comma after “In neonates”
Row 99 – change have to has
Row 106 – rephrase “ 460 infants with SVT proposed Wolff-”
References:
You need to update some of them and adjust all of them according to journal’s specifications.
Author Response
Thank you for your thoughtful review of our manuscript. Your comments and suggestions have been incorporated into the revised draft.
Reviewer 2 Report
The authors intended to provide evidence of the effectiveness of electric countershock therapy for neonatal arrhythmia by conducting this systematic review. However, I have one primary concern plus one minor comment.
1. As the author described, most included studies were case reports and case series. There could be significant reporting bias since we all know that success cases are more easily to be published. Conversely, failed treatments were seldom reported. In this case, I doubt that the conclusion of this study could not represent the actual figure clinically.
2. I recommend not using "wrong" in the flow diagram. In addition, each aspect of the reason for excluding studies in the flow diagram should be defined in detail as the inclusion criteria in the Methods. For example, exposure, language, study design, intervention...should all be defined in detail in the Methods under the subheading of "Inclusion and exclusion criteria".
Author Response
1. As the author described, most included studies were case reports and case series. There could be significant reporting bias since we all know that success cases are more easily to be published. Conversely, failed treatments were seldom reported. In this case, I doubt that the conclusion of this study could not represent the actual figure clinically.
Answer: We thank the Reviewer for the important hint on the limitations of this review and added a subheading where we point out this possible reporting bias.
2. I recommend not using "wrong" in the flow diagram. In addition, each aspect of the reason for excluding studies in the flow diagram should be defined in detail as the inclusion criteria in the Methods. For example, exposure, language, study design, intervention...should all be defined in detail in the Methods under the subheading of "Inclusion and exclusion criteria"
Answer: We changed the prisma chart details, so reasons for excluded studies are clearer defined.
Reviewer 3 Report
I had reviewed this manuscript on “Cardiac Arrhythmias Requiring Electric Countershock during the Neonatal Period: A Systematic Review”.
Minor comments on the table, suggest to put the title of the table 1&2 on the top. For table 1, suggest to put publication/authors at first column and number of patients after study design.
Author Response
Minor comments on the table, suggest to put the title of the table 1&2 on the top. For table 1, suggest to put publication/authors at first column and number of patients after study design.
Answer: We thank the Reviewer for the suggestions on the table. We corrected the table in your mentioned arrangement.
Reviewer 4 Report
This is a very interesting and well-documented systematic review. The few issues that need further improvement are as follows:
- First, I would use the term ”electric cardioversion”, to differentiate it from pharmacological cardioversion
- Tha paragraph from row 55 to 63 should be rewritten – it feels like a bunch of phrases with almost no logical connection
- On row 139, the authors should nonetheless mention the percentages of sexes affected by arrhythmias
- Before mentioning the type of electric countershock (row 180), the authors should differentiate between the studies in which infants received some kind of anti-arrhythmic medication prior to being subjected to electric therapy – and this mandates further the discussion whether in some settings, the electric shock therapy is performed due to the unavailability of specific medications…
- On page 16, row 96 (?) Neonatal Resuscitation is mentioned, but it should be differentiated from any resuscitation performed in the neonatal period: the Neonatal Resuscitation as it is described in the guidelines, only refers to the resuscitation performed in the delivery/operating room at birth, when there is a troublesome transition to the extrauterine environment, there are no known arrhythmias and only the potential cardiac arrest is in question – I think this kind of differentiation should be discussed.
Author Response
First, I would use the term ”electric cardioversion”, to differentiate it from pharmacological cardioversion
- Tha paragraph from row 55 to 63 should be rewritten – it feels like a bunch of phrases with almost no logical connection
- On row 139, the authors should nonetheless mention the percentages of sexes affected by arrhythmias
- Before mentioning the type of electric countershock (row 180), the authors should differentiate between the studies in which infants received some kind of anti-arrhythmic medication prior to being subjected to electric therapy – and this mandates further the discussion whether in some settings, the electric shock therapy is performed due to the unavailability of specific medications…
- On page 16, row 96 (?) Neonatal Resuscitation is mentioned, but it should be differentiated from any resuscitation performed in the neonatal period: the Neonatal Resuscitation as it is described in the guidelines, only refers to the resuscitation performed in the delivery/operating room at birth, when there is a troublesome transition to the extrauterine environment, there are no known arrhythmias and only the potential cardiac arrest is in question – I think this kind of differentiation should be discussed.
Answer: We thank the Reviewer for the important feedback on our review. We totally agree with the term “electric cardioversion” for clearer definition. Also, as you suggest we differentiate more between studies where neonates received electric countershock with or without an antiarrhythmic premedication.
We totally agree to differentiate more between neonatal resuscitation guidelines and advanced pediatric live support, we added this into the discussion section.
Reviewer 5 Report
Thsi is an interesting article assessing arrhythmias occurring in the peri- and post-natal period and being treated by electric countershock. I believe that the article is well written and results are sound. However is should be considered as a covariate presence of prenatal diagnosis, severity and treatment carried out during fetal life. A recent review provided a practical guide for the diagnosis and management of common fetal arrythmias, from the joint perspective of the fetal medicine specialist and the cardiologist. This evidence should be considered and a brief sentence added along with the citation as the multidisciplinary assessment is a milestone for fetal arrhithmias (ref 1). Neonatal severity of the condition is a function of prenatal course and management and this cannot be ingnored as it may be a major predict of survival and success of treatment (this includes: timely referral to center with expertise, timely delivery, adequate way of delivery, etc).
Please emphasize a bit more neonatal age at treatment. If the neonate is liveborn unaffected and subsequently develops the arrhithmya we may anticipate a rapid deterioration but we may be certain that there is no hydrops of cardiac insufficiency. On the other hand if the newborn is affected at birth generally the condition has a longer natural hystory with intrauterine onset and the risk of hydrops is higher (and neonatal demise/unsuccesful treatment). Please try to emphasize more that these are different populations with differet risks to some extent and some subanalyses should be required on this end.
Author Response
Thsi is an interesting article assessing arrhythmias occurring in the peri- and post-natal period and being treated by electric countershock. I believe that the article is well written and results are sound. However is should be considered as a covariate presence of prenatal diagnosis, severity and treatment carried out during fetal life. A recent review provided a practical guide for the diagnosis and management of common fetal arrythmias, from the joint perspective of the fetal medicine specialist and the cardiologist. This evidence should be considered and a brief sentence added along with the citation as the multidisciplinary assessment is a milestone for fetal arrhithmias (ref 1). Neonatal severity of the condition is a function of prenatal course and management and this cannot be ingnored as it may be a major predict of survival and success of treatment (this includes: timely referral to center with expertise, timely delivery, adequate way of delivery, etc).Please emphasize a bit more neonatal age at treatment. If the neonate is liveborn unaffected and subsequently develops the arrhithmya we may anticipate a rapid deterioration but we may be certain that there is no hydrops of cardiac insufficiency. On the other hand if the newborn is affected at birth generally the condition has a longer natural hystory with intrauterine onset and the risk of hydrops is higher (and neonatal demise/unsuccesful treatment). Please try to emphasize more that these are different populations with differet risks to some extent and some subanalyses should be required on this end.
Answer: We thank the Reviewer for the important feedback in management of fetal arrhythmias and the great influence of postnatal mortality and morbidity. We tried to highlight this in the discussion section.